# Problem-Solving Skills Training for Parents of Children Undergoing Hematopoietic Stem Cell Transplantation: A Mixed Methods Feasibility Study

**DOI:** 10.3390/cancers17060930

**Published:** 2025-03-10

**Authors:** Heather Bemis, Mikela Ritter, Maxwell (Nina) Lee, Paula Murray, Robert Noll, Rebecca Barber, Chelsea Balian, Jessica Ward

**Affiliations:** 1Division of Comfort and Palliative Care, Department of Anesthesiology Critical Care Medicine, Children’s Hospital Los Angeles, Los Angeles, CA 90027, USA; mdritter@uw.edu (M.R.); nilee@chla.usc.edu (M.L.); 2Department of Pediatrics, Keck School of Medicine, University of Southern California, Los Angeles, CA 90027, USA; 3Institute for Nursing and Interprofessional Research, Children’s Hospital Los Angeles, Los Angeles, CA 90027, USA; paula.murray@sickkids.ca (P.M.); rbarber@chla.usc.edu (R.B.); jward@chla.usc.edu (J.W.); 4Department of Pediatrics, University of Pittsburgh, Pittsburgh, PA 15260, USA; rbn1@pitt.edu; 5Department of Stem Cell Transplantation and Cellular Therapy, Children’s Hospital Los Angeles, Los Angeles, CA 90027, USA; cbalian@chla.usc.edu

**Keywords:** pediatric stem cell transplantation, parent psychological distress, problem-solving skill training

## Abstract

Caring for a child receiving hematopoietic stem cell transplantation (HSCT) can be stressful. This study tested Bright IDEAS^®^, a problem-solving skills program, to see if it is feasible and acceptable for caregivers in the HSCT setting. Caregivers were assigned by chance to receive Bright IDEAS^®^ with usual care or usual care alone. Bright IDEAS^®^ involved six-to-eight sessions to empower caregivers to manage challenges. IDEAS stands for I—identify problems, D—define options, E—evaluate pros and cons, A—act with a plan, and S—see if it worked. Most caregivers assigned to Bright IDEAS^®^ completed the program and found it helpful. Interviews showed that caregivers appreciated the program’s flexibility and support. Overall, the study results suggest that Bright IDEAS^®^ is a promising way to help caregivers during their child’s HSCT, and it may decrease symptoms of distress, but needs to be tested in a larger study.

## 1. Introduction

Approximately 6200 allogeneic and 560 autologous hematopoietic stem cell transplantation (HSCT) procedures were performed for children and adolescents aged <18 years between 2018 and 2022 in the United States [1]. HSCT is a curative therapy for children and adolescents with serious, potentially life-limiting illness, including high-risk, refractory, or recurrent malignancies, immunodeficiencies, and hematologic disorders. However, HSCT procedures are associated with compromised physical, psychological, and social wellbeing [2]. Common medical complications of HSCT procedures include infection, pain, graft failure, and graft-versus-host disease, among other distressing symptoms. Although symptom management strategies are implemented, children undergoing HSCT continue to report a high symptom burden at the time of cell infusion and through the 90 days post HSCT [3].

Alongside their child’s intensive treatment, caregivers of children undergoing HSCT face significant sources of stress, including prolonged and repeated child hospitalizations, engaging in complex medical care for their child, disruptions to family routines, and the threat of the possible death of their child. Caregivers in this population are at increased risk of psychological distress [4], with one study reflecting elevated rates of suicidal ideation, ranging from 27% to 39% of caregivers across the time period from stem cell infusion to 90 days later [5]. Caregiver distress has also shown to be significantly associated with poor child health-related quality of life and an increased symptom burden across this time period [6].

Evidence-based interventions are needed to reduce caregiver distress and improve child outcomes for children undergoing HSCT, yet few have been adapted for this population, partly due to the complexities of implementation in this intensive setting. The present study is a pilot randomized controlled trial to test the feasibility and acceptability of delivering an evidence-based cognitive–behavioral intervention in this underserved, high-need setting. Psychoeducational interventions (PEIs) for parents of sick children aim to reduce adverse psychological outcomes, decrease maladaptive parenting behaviors, improve family functioning, and promote the child’s health and wellbeing [7]. A recent meta-analysis of randomized controlled trials (RCTs) concluded that PEIs can lead to significant reductions in symptoms of post-traumatic stress, improved mood, and acquisition of problem-solving skills in caregivers of children with cancer [8]. Problem-solving skills training (PSST), specifically, yielded superior effects on caregiver distress and child outcomes for caregivers of children with serious illnesses when compared to other PEIs [9].

PSST is based on the social problem-solving model and emphasizes the importance of fostering enhanced problem-solving skills to support constructive problem-solving and subsequent reduction in emotional distress [9]. Studies of PSST for caregivers have demonstrated improvements in quality of life (QoL) in children with asthma [10], have reduced negative affectivity and distress amongst mothers of children newly diagnosed with cancer [11,12], and have decreased distress in caregivers of adults undergoing HSCT [13,14].

To date, PSST interventions have not been studied in caregivers of children undergoing HSCT. Other caregiver-directed PEIs studied in the pediatric HSCT setting such as massage with relaxation training and guided imagery [15] or stress-reduction education sessions [16] have yielded minimal beneficial effects and documented high drop-out rates with significant missing data. A social–cognitive processing intervention for caregivers of children undergoing HSCT reported an attrition rate of approximately 30% over the course of the intervention, with beneficial effects on caregiver distress that did not endure beyond the short term [17].

Bright IDEAS^®^ (BI) is a cognitive–behavioral PSST intervention with demonstrated efficacy for caregivers of children newly diagnosed with cancer [12], but has not been trialed in the unique, intensive HSCT setting. BI applies a specific method of problem-solving [11,12], where Bright is optimism and IDEAS mark the steps of problem-solving (Identify the problem, Define your options, Evaluate the pros and cons of each option, Act with a specific action plan, and See if it worked).

The primary objective of this research study is to determine the feasibility and acceptability of BI for caregivers of children undergoing HSCT, using a mixed methods approach within a socioeconomic and ethnically diverse population. The secondary objective is to evaluate the preliminary effectiveness of BI in improving problem-solving skills and reducing psychological distress compared to a control group of caregivers receiving the usual psychosocial care. By integrating a mixed methods approach to evaluate feasibility and acceptability, caregivers’ voices are incorporated. Caregiver qualitative feedback may be utilized in future iterations of the intervention to refine content, features, and procedures.

## 2. Methods

### 2.1. Design

This study employed a randomized, controlled pilot design to test the feasibility, acceptability, and preliminary effect of Bright IDEAS^®^ among primary caregivers of children and adolescents undergoing autologous or allogeneic HSCT with myeloablative or reduced intensity conditioning. The six-to-eight-session intervention and data collection started upon admission to the HSCT inpatient unit, after randomization to intervention or control condition. Repeated psychological outcome measures were collected from caregivers at baseline (prior to their child’s stem cell infusion), then 60, 90, and 180 days after cell infusion (four time points in total; Figure 1).

### 2.2. Setting and Sample

Using a convenience sampling method, participants were recruited from the Transplantation and Cellular Therapy Program at a large urban children’s hospital where approximately 100 allogeneic and autologous HSCTs are performed annually. The target sample size was 50 participants, randomized 1:1 to the Bright IDEAS^®^ or control (usual care) group. A formal power analysis was not conducted; rather, the target sample size was estimated to provide sufficient feasibility and preliminary effect data. A 12-month recruitment period was planned. Caregivers of male or female children and adolescents of any race or ethnicity, aged 2 to 21 years, and scheduled to receive allogenic or autologous HSCT using myeloablative (high-intensity chemotherapy, with or without total body irradiation) or reduced-intensity chemotherapy conditioning were eligible to participate. One English- or Spanish-speaking primary caregiver per child was eligible to participate. Caregivers were excluded if they (or their child) were concurrently enrolled on a research study testing psychoeducational interventions for caregivers and/or patients.

### 2.3. Procedures

Institutional review board approval was obtained. Eligible caregiver–child dyads were identified from the transplantation referral list. Caregivers and children were invited to participate in the study after the child was hospitalized for the HSCT admission, prior to the initiation of reduced intensity or myeloablative conditioning and stem cell infusion. English and Spanish consent and assent forms were available. Caregiver informed consent and permission for child participation was obtained for all study participants. Assent was obtained as per institutional guidelines. Caregiver and children baseline screening data were collected at the time of study enrollment.

Stratified randomization, using the REDCap^®^ (Nashville, TN, USA) randomization module [18], was performed to ensure that an equivalent number of autologous and allogeneic HSCT dyads were assigned to the intervention and control groups. The intervention was initiated after baseline measures were completed. Caregivers completed psychological assessment measures electronically via links sent to their computer, smart phone, or tablet, using REDCap^®^, regardless of their physical location (inpatient unit, clinic, home). Trained study personnel were available to answer questions or provide support as caregivers completed measures at each timepoint. If electronic measure completion was not possible or preferred by the caregiver, paper measures were provided.

Deidentified child demographic, underlying diagnosis (indication for HSCT), and HSCT data were extracted from the electronic medical record and entered into REDCap^®^ case report forms by study ID number. Caregivers were eligible to receive up to USD 100 in gift cards for measure completion (USD 20 at baseline, USD 30 after 3 months, and USD 50 after 6 months). A subset of caregivers who completed the intervention were invited to participate in semi-structured interviews. Those who completed the interview received an additional USD 50.

### 2.4. Intervention and Control Condition

The study intervention, Bright IDEAS^®^, is a problem-solving skills training program (Figure 1) developed by physicians and psychologists working in pediatric oncology [19]. Bright IDEAS^®^ materials are available in English and Spanish in the public domain at https://ebccp.cancercontrol.cancer.gov/programDetails.do?programId=546012 (accessed on 3 March 2025). Consistent with the established Bright IDEAS^®^ manual [12], intervention participants attended up to eight individual, face-to-face sessions with a trained BI interventionist over eight weeks. Sessions were scheduled weekly, with flexibility to participant needs. Sessions took place at bedside or in a private room in the transplant unit, per availability and participant preference. Telehealth sessions were offered as needed based on caregivers’ preference during hospitalization, as well as after discharge. Caregivers were considered to have completed the intervention if they participated in six sessions [12], with the option to complete up to eight determined by participant interest and progress.

Intervention sessions and all BI materials were available in English and Spanish [11,12]. Sessions entailed the use of worksheets teaching a stepwise process to assist and empower caregivers to contemplate and document challenges and goals in their life (worksheet #1), focus on and identify elements of a single problem to solve (worksheet #2), define and evaluate potential solutions (worksheet #3), and trial an action plan with the opportunity to review the impact and need for revision of the plan (worksheet #4; Table 1). Caregivers were prompted to identify one caregiver-level problem, such as relational, logistical, or financial, and one child-level problem that related to a distressing or persisting symptom the child was experiencing. Bright IDEAS^®^ was individualized to address specific problems identified by the participating caregivers. Spanish-speaking interventionists or Spanish interpreters (in-person, via telephone, or via iPad when bilingual interventionists were not available) were used to conduct intervention sessions for Spanish-speaking caregivers.

Usual care was used as the control condition, offered to all study participants, including those randomized to Bright IDEAS^®^, and entailed pre-HSCT assessments done by a dedicated HSCT-trained social worker followed by ongoing social work support, as needed, for the duration of the child’s transplantation journey. In addition, psychologists and psychiatrists were available on a consultative basis for children and adolescents in need, but did not provide direct services for caregivers. All study participants were informed about available psychosocial resources for patients and families. Comparison to usual care was used to evaluate the effects of Bright IDEAS^®^ as an adjunctive intervention.

Interventionists on the study team received Bright IDEAS^®^ training prior to engaging in sessions with caregivers. Training materials and procedures were developed and standardized by members of the Bright IDEAS^®^ leadership team who have trained over 300 professionals. Training for the study team consisted of didactics, case-based scenarios, and mentored role-play experiences.

### 2.5. Baseline and Psychological Outcome Measures

Caregiver outcomes were assessed at baseline (T1, prior to stem cell infusion and BI intervention), then 60 days (T2), 90 days (T3), and 180 days (T4) post stem cell infusion (four time points in total). The Caregiver Demographic and Health Questionnaire (previously described) was used to measure parent sociodemographic and health information [20].

Caregiver problem-solving skills were evaluated using the Social Problem-Solving Skills Inventory Revised Short Form, a 25-item tool that measures five dimensions of problem-solving using a five-point Likert scale [21]. Higher scores indicate better problem-solving skills. Strong reliability and validity have been documented in similar samples of caregivers caring for children with serious illnesses [22].

The Beck Anxiety Inventory (BAI)-I was used to measure caregiver anxiety. The BAI-I is a 21-item, multiple choice, self-report instrument of adult anxiety symptoms (somatic, subjective, and panic-related) [23]. A four-point Likert-type scale structure is used. Item responses range in intensity from 0 (not at all) to 3 (severe). Previous use of the BAI-I in a study involving outpatient adults with psychiatric conditions demonstrated an internal reliability coefficient of 0.92 [23]. Construct validity of the BAI-I has been supported by its convergence with other validated measures of anxiety in studies of adults with anxiety disorders [24].

The Beck Depression Inventory (BDI)-II, a 21-item, multiple choice, self-report measure [25], was used to measure depressive symptoms. A four-point Likert-type response structure is used with items ranging in intensity from 0 (not at all) to 3 (severe). An internal consistency of approximately 0.93 was demonstrated in a prior study of adults with depression, treated in inpatient and outpatient settings [26].

Item #9 of the BDI-II evaluates suicidal ideation. The response options are as follows: (1) I don’t have any thoughts of killing myself, (2) I have thoughts of killing myself, but I would not carry them out, (3) I would like to kill myself, (4) I would kill myself if I had the chance. A standard operating procedure (SOP) for the BDI suicide item was in place for the duration of this study, as previously described [5].

The Impact of Events Scale Revised consists of 22 items that uses Likert-type responses to measure perceived post-traumatic stress and has acceptable psychometrics [27]. The instrument is organized into three subscales (intrusion, avoidance, and hyperarousal). Higher scores indicate increased severity of distress and more symptoms of post-traumatic stress.

The Profile of Mood States (POMS) Short Form, a 30-item self-report instrument with excellent psychometric properties [28], was used to measure negative affectivity. This tool includes five-point, Likert-type items and seven subscales (tension/anxiety, depression/dejection, anger/hostility, fatigue, confusion, vigor, and friendliness). Higher scores represent a greater mood disturbance.

### 2.6. Qualitative Acceptability

Semi-structured acceptability interviews were performed to detect themes that represent participant perception of the intervention. To meet thematic saturation within time and budgeting considerations, fifty percent of the intervention completers (*N* = 7) were offered to participate in the semi-structured interviews. Interviewees were representative of the full BI group with respect to language, race, ethnicity, gender, and child age. The duration of each interview was approximately 30–45 min. An interview guide was used and audio recordings were transcribed. All interviews were conducted by a member of the study team unfamiliar to the participant (i.e., not their interventionist).

### 2.7. Analyses

Quantitative analyses included descriptive statistics to summarize the feasibility of the intervention as assessed by enrollment rates, withdrawal rates, intervention completion, and survey completion. Rates of missing data and attrition were analyzed. Caregiver psychological outcome measures were scored as per the published scoring instructions. Central tendency analyses were performed for sociodemographic data, child HSCT data, and caregiver psychological outcomes. Anxiety and depression scores were evaluated based on clinical cutoffs for each severity level. All data analyses were conducted using the R software version 4.4.2 [29].

To execute the qualitative analyses, all interviews were digitally recorded and transcribed verbatim by a professional transcription service, including professional translation for interviews conducted in Spanish. Three members of the research team performed a thematic analysis [30] of the interview transcripts. Data were coded deductively using the interview guide for overall satisfaction, perceived change, and planned skill use, as well as inductively using concepts introduced by participants regarding their experience. Coders independently reviewed transcripts to identify initial codes, then met to resolve discrepancies and reach a consensus on operationally defined codes that were entered into a formal codebook. Dual coders then applied the coding framework in a second cycle to all interview transcripts, and the team subsequently met to reach a consensus on final themes.

## 3. Results

### 3.1. Recruitment, Enrollment and Retention

One-hundred seven dyads were screened and 64 were eligible and approached by the study team (Figure 2). Thirty-eight dyads were enrolled and randomized (59.4%); a total of 20 were randomized to the BI intervention arm and 18 were randomized to the control arm (Figure 2). Primary reasons for caregiver exclusion were prior patient enrollment in a conflicting research study (*N* = 12, 27.9%), language criteria for survey completion (*N* = 5, 11.6%), and study staff inability to contact the caregiver within the approximately 10-day pre-transplant enrollment window (time from admission to the HSCT unit and duration of conditioning to cell infusion; *N* = 15, 34.8%). Of the 26 who declined participation, 10 (38.5%) indicated not having time or being too overwhelmed, while 16 (61.5%) were uninterested in the study and/or research in general.

Of the 38 dyads enrolled, seven participants later withdrew from the study (18.4%), comprising six withdrawals from the BI group and one from the control group (Figure 2). Two caregivers were withdrawn for failing to complete the baseline surveys, including the caregiver randomized to the control group, one caregiver was withdrawn due to lack of sufficient reading comprehension in English or Spanish, two caregivers withdrew due to feeling overwhelmed and not having enough time to participate, and two caregivers were lost to the follow-up.

Of the 20 caregivers randomized to Bright IDEAS^®^, two were withdrawn before any sessions could be completed (due to difficulty with reading comprehension or failure to complete baseline surveys, as above). Of the remaining 18 participants, 14 (77.8%) completed the intervention (≥six sessions). Participants attended an average of 5.7 sessions, with 16 caregivers attending at least two. The primary reason for not completing the intervention was caregiver time constraint. The majority (76.6%) of sessions took place in person.

In the BI group, questionnaire completion rates across the four follow-up time points were as follows: 90% (baseline), 89.5% (day 60), 84.2% (day 90), 84.2% (day 180). In the control group, questionnaire completion rates were: 94.4% (baseline), 100% (day 60), 94.4% (day 90), 77.8% (day 180).

### 3.2. Sample Characteristics

The majority of children enrolled had an underlying diagnosis of cancer (85% in the BI group, 55.5% in the control group) and received an allogeneic HSCT (55% in the BI group, 66.7% in the control group) (Table 2). All caregivers in the BI group were female; two caregivers (11.1%) in the control group were males. The majority of the caregivers were white (40% in the BI group, 33.3% in the control group) and spoke English (65% in the BI group, 77.8% in the control group). The participant sub-sample for qualitative interviews (*N* = 7) was similar to the overall BI group with respect to demographic characteristics; however, interviewees’ children all received malignancy diagnoses and allogenic transplants (vs. 85% and 55%, respectively, in the full BI group).

### 3.3. Quantitative, Psychological Outcome Measures

Preliminary efficacy analyses probed for changes to self-reported psychological outcome measures across time among participants in the intervention group versus the control, following established guidelines on efficacy reporting in behavioral feasibility studies that are not powered to detect statistically significant changes [31]. Table 3 presents the mean (standard deviation) of caregiver psychological outcome measures over time. Anxiety and depression scores at baseline were higher in the BI group compared to caregivers in the control group. A downtrend in anxiety and depression scores was observed at the follow-up timepoints in the BI group. Impact of Events and Profile of Mood States scores were similar between groups at baseline. These scores remained stable in the BI group, with a downtrend noted in the control group at the follow-up timepoints. Problem-solving skill scores were similar between groups at baseline and remained stable over time.

Table 4 displays caregiver anxiety and depression scores based on severity categorization determined by clinical cutoffs. Moderate and severe anxiety (Table 4a) was observed in the BI and control groups at baseline; no participants in the BI group reported moderate or severe anxiety on day 90 or 180. Fewer caregivers in the BI group reported moderate depression on day 90 and 180 compared to baseline (Table 4b). No caregivers in the BI group reported severe depression on day 180.

Suicidal ideation was also assessed as a proxy for caregiver distress, using responses to BDI item #9. Responses of 1 or greater were considered to be indicative of suicidal ideation. In the BI group, suicidal ideation was as follows: pre-infusion: *N* = 1 (5.6%); day 60: *N* = 1 (8.3%); day 90: *N* = 0 (0.0%); day 180: *N* = 0 (0.0%). The frequency of suicidal ideation in the control group was as follows: pre-infusion: *N* = 0 (0.0%); day 60: *N* = 2 (18.2%); day 90: *N* = 0 (0.0%); day 180: *N* = 0 (0.0%). Frequencies were calculated based on the total number of participants who completed BDI #9 at each time point. No caregivers endorsed responses of 2 or 3 at any timepoint.

### 3.4. Qualitative, Semi-Structured Interviews

Participant feedback from semi-structured interviews supported acceptability to caregivers, with 100% (*N* = 7/7) of those interviewed endorsing overall satisfaction with their experience, perceived positive change in wellbeing following the intervention, and plans to continue using BI skills.

A thematic analysis of qualitative interviews further evaluated caregivers’ appraisals of feasibility, implementation, and acceptability (Table 5). An unexpected theme emerged, indicating that BI exceeded participant expectations, as over half (*N* = 4/7) of the interviewees expressed unprompted appreciation or gladness to have participated in BI despite initial reticence to enroll in the study. Similarly to enrollment barriers, participants indicated that the primary challenges to session participation included time and competing medical needs (e.g., interruptions for routine medical cares, meetings or appointments with other providers), while responsiveness to scheduling needs facilitated completion. Relatedly, participants expressed mixed preferences regarding timing and length of the intervention and appreciated that BI provided flexibility and individual choice around scheduling. Most interviewees (*N* = 5/7) expressed a preference for in-person participation versus telehealth, while two were uncertain or preferred a mix of both formats. Participants indicated that they found the one-to-one interventionist role helpful in providing supportive listening and guidance or accountability in using the BI model.

## 4. Discussion

### 4.1. Summary and Synthesis of Findings

The goal of the current study was to examine the feasibility and acceptability of delivering BI to caregivers of children undergoing HSCT. This randomized, controlled pilot represents a novel application of an evidence-based, structured cognitive–behavioral intervention to improve problem-solving skills and alleviate psychological distress within this uniquely intensive setting in which caregiver support is highly needed yet potentially difficult to access [32].

Results of this BI pilot suggest that the intervention was feasible and acceptable for caregivers of children undergoing HSCT. A priori targets were not set given the dearth of existing PEI studies in pediatric HSCT; however, rates of enrollment (59.4%), retention (81.2%), and survey completion (89.3% across timepoints) were slightly favorable compared to other psychosocial RCTs involving patient–caregiver dyads in cancer populations [33]. Among the caregivers randomized to BI, the majority (70%) completed the intervention. Participant withdrawal from the study was primarily due to time constraints. No participants cited dissatisfaction with BI as a reason for withdrawal.

The current study focuses on feasibility and acceptability of BI in the HSCT setting and did not incorporate formal hypothesis testing of intervention effectiveness [31]. However, trends across timepoints cautiously suggest reductions in symptoms of depression and anxiety in the BI group versus the usual care control with stable follow-up effects. Initial changes in post-traumatic stress symptoms and negative mood were greatest at the first post-intervention follow-up. Notably, the intervention group demonstrated higher levels of psychological distress at baseline, suggesting that BI may be most impactful among caregivers with elevated symptoms of distress. This would be consistent with recent literature suggesting that a resilience skill-building intervention for adolescents and young adults receiving HSCT would be most beneficial for participants with preexisting anxiety and depression [34]. Other recent studies have indicated a risk of caregiver distress not only at the time of HSCT transplant but longitudinally [5,35], while caregivers who utilize approach-oriented coping strategies demonstrate lower distress in adult HSCT populations [36].

All the caregivers interviewed reported high satisfaction with BI, including a perceived positive experience, change, and durability of skill engagement. Though participants appreciated the flexibility afforded by the option to conduct sessions via telehealth, and despite prior research demonstrating the effectiveness of web-based delivery of BI [37], most interviewees in the current study reflected a greater sense of subjective benefit from in-person sessions. Interviewees reflected upon an initial reticence to enroll due to competing priorities at the time of recruitment, alongside gladness or gratitude for having completed BI.

Qualitative and quantitative findings suggest that caregivers benefited from participating in BI. This finding is similar to recent intervention studies involving caregivers of adult patients facing HSCT or advanced cancer [38,39]. Utilizing a “traffic light” model to evaluate feasibility [40], the present results point to an “amber” status, highlighting potential need for revision to the recruitment/engagement approach to support upfront intervention accessibility within the intensive setting of pediatric HSCT. One such strategy could entail expanding the initial engagement window to allow for greater flexibility and responsiveness to individual needs regarding the timing of the intervention.

The results of this pilot trial are consistent with prior studies that have demonstrated the feasibility and acceptability of BI among caregivers of children recently diagnosed with cancer [12]. Recent caregiver interventions with demonstrated feasibility in the HSCT setting have focused on caregiver wellbeing via a positive psychology intervention among caregivers of HSCT survivors [41] and via music play experiences in a small sample of caregivers of young children [42].

### 4.2. Strengths and Limitations

The present study has several strengths that positively impact its generalizability and support the case for a larger, statistically powered efficacy trial. First, this study included a diverse sample of both English- and Spanish-speaking caregivers. Second, the current pilot is the first to use a mixed methods design incorporating validated quantitative outcome measures of caregiver anxiety, depression, post-traumatic stress symptoms, and negative affectivity in addition to qualitative semi-structured interviews to evaluate a PSST intervention in the HSCT setting. Third, the employment of a narrow pre-transplant recruitment window, a pre-transplant baseline assessment, and multiple follow-up time points allowed for a controlled, longitudinal examination of the feasibility of a BI trial during the most intensive window of HSCT treatment.

This study also included several limitations that are important to consider when interpreting the results and that provide additional direction for future research. Though qualitative satisfaction with BI was high, it is possible that this could reflect selection bias, as interviews involved a subset of caregivers who completed at least six BI sessions. Intervention completers may have been more likely to be satisfied with the intervention than non-completers. While interviewees matched the broader BI cohort with respect to demographics, they comprised all parents whose child had a malignancy diagnosis and allogenic transplant, and it cannot be ruled out that their experience differed due to factors related to their child’s diagnosis or treatment. Although no non-completers cited dissatisfaction as a reason for withdrawal, a larger trial could strengthen this evaluation by attempting to include standardized satisfaction assessments for non-completers. Conversely, though feasibility metrics were acceptable for this trial, it is possible that the rates reported underestimate the enrollment and retention potential in this population due to barriers impacting eligibility rates, staffing, and patient access and staff availability during the COVID-19 pandemic. Further, male caregivers were underrepresented in this study and may be particularly at risk of emotional challenges following their child’s HSCT [35].

This pilot trial was not powered to assess efficacy of the intervention nor potential mediators or moderators of effect using statistical significance testing and did not meet the target enrollment of 50 participants. Despite reductions in psychological distress being observed descriptively in the BI group versus the control group, minimal differences were noted in participants’ self-reported problem-solving skills. Therefore, it cannot be determined whether changes observed in the BI group were due to the intervention or extraneous variance related to other factors. Previous work has demonstrated that improved problem-solving skills mediated less distress in caregivers of children with cancer [12]. Further, more children of caregivers in the BI group had a diagnosis of malignancy compared to children of caregivers in the control group. This observation may contribute to the higher baseline anxiety and depression reported by caregivers in the BI group.

### 4.3. Future Research

An in-depth analysis of the rich qualitative data obtained in the study is needed to better understand elements of the BI intervention that influence acquisition and maintenance of problem-solving skills in this caregiver population. Additionally, further research employing a randomized controlled design with a larger sample is needed to rigorously examine the short- and long-term effectiveness of BI and the mechanism of the potential effect on caregiver and child outcomes in the HSCT setting. Future research should consider incorporating rigorous examinations of potential moderators to best predict BI’s effectiveness; for example, utilizing an a priori cutoff score on baseline measures of distress to guide participant eligibility and randomization procedures might assist in further illuminating which caregivers would benefit most from this resource. Further, prospective and objective measures of caregiver stress and psychosocial support should be added for a comprehensive assessment of caregiver psychological status.

The inclusion of child diagnosis, transplant course, and health care utilization cofactors will be important in future trials with respect to both quantitative and qualitative outcomes. Transplant type and complications such as infection, graft-versus-host disease, graft failure, veno-occlusive disease, post-HSCT disease recurrence, and survival are such examples. Hospital length of stay, intensive care unit transfer, emergency department visits, and readmission may play a role in post-HSCT caregiver distress and may impact BI utilization and effectiveness over time. Measurements of child symptom burden and health-related quality of life should be included in future studies, as these constructs may influence caregiver psychological distress and effectiveness of BI.

## 5. Conclusions

This study is the first to examine the feasibility and acceptability of Bright IDEAS^®^, an evidence-based problem-solving skills intervention, to address psychological distress in caregivers of children undergoing HSCT. A rigorous, mixed-method approach was utilized, including a randomized and longitudinal design, semi-structured interviews, and validated psychological measures to pilot BI with a diverse sample of English- and Spanish-speaking caregivers. The results indicate that Bright IDEAS^®^ is feasible and acceptable to caregivers in this setting; therefore, further studies among larger, more balanced samples within multiple HSCT settings are warranted to ensure the intervention’s generalizability across diverse demographic groups. In particular, including greater representation of fathers will help provide a more comprehensive understanding of how BI impacts various caregiver subgroups and identify potential variations in effectiveness. Future randomized controlled trials should also consider including greater flexibility in the recruitment window. These adjustments will allow for increased access, understanding of BI’s effectiveness, and factors that might impact its effect in order to best tailor this intervention to patient needs in this intensive, underserved setting.

## Figures and Tables

**Figure 1 cancers-17-00930-f001:**
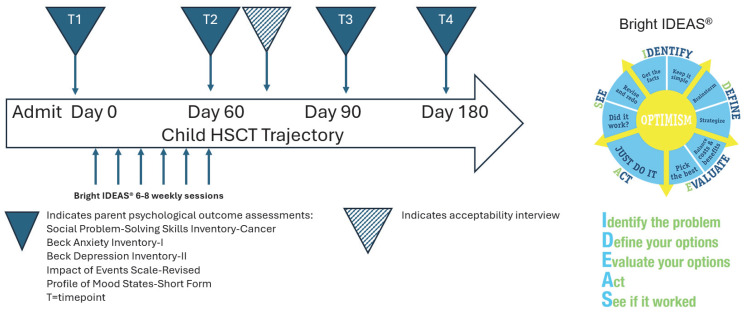
Study schema.

**Figure 2 cancers-17-00930-f002:**
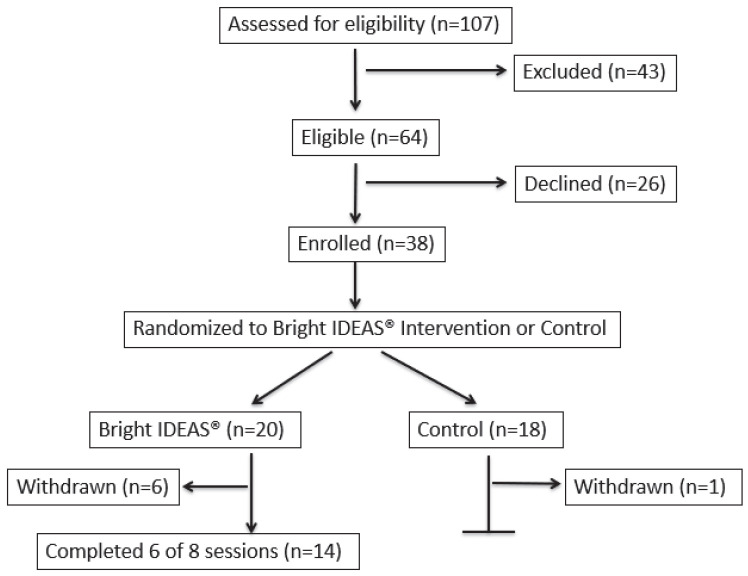
Study enrollment and randomization consort diagram.

**Table 1 cancers-17-00930-t001:** Bright IDEAS^®^ session structure.

Session	Content	Materials
1	Establishing rapport; learning about the family’s medical journey Explaining the Bright IDAS model Identifying current challenges/problems	Provider manual User manual and brochure Worksheet #1
2–5	Teaching and practice of the BI steps Identifying the core elements of the chosen problem or challenge Defining and evaluating the pros and cons of potential solutions Creating an action plan to implement the chosen solution(s) See/assess if the plan worked, revise as needed	Provider and user manuals Worksheet #2 Worksheet #3 Worksheet #4 Worksheet #4
6–8	Review of the BI steps and continued practice for mastery Applying steps to additional chosen challenge as needed	Provider and user manuals Worksheets #2–4

**Table 2 cancers-17-00930-t002:** Sample characteristics (*N* = 3.8).

		Mean ± SD/*N* (%)	
		Bright Ideas (*N* = 20)	Control (*N* = 18)
Child Demographic, Diagnosis, and Therapy			
Age	Child	6.0 ± 5.3	6.8 ± 5.0
Gender	Male Female	8 (40.0) 12 (60.0)	10 (55.6) 8 (44.4)
Diagnosis	Malignancy Red cell disorder Immune deficiency Metabolic disorder Other	17 (85.0) 0 (0.0) 3 (15.0) 0 (0.0) 0 (0.0)	10 (55.5) 3 (16.7) 2 (11.1) 1 (5.6) 2 (11.1)
Treatment Type	Allogeneic Autologous	11 (55.0) 9 (45.0)	12 (66.7) 6 (33.3)
Parent Self-Report Demographic Information			
Age	Parent	37.6 ± 7.3	36.5 ± 6.7
Gender	Male Female	0 (0.0) 20 (100.0)	2 (11.1) 16 (88.9)
Race *	White African-American Asian Native Hawaiian or other Pacific Islander American-Indian or Alaska Native Other Do not wish to answer Missing	8 (40.0) 0 (0.0) 0 (0.0) 0 (0.0) 0 (0.0) 3 (15.0) 5 (25.0) 4 (20.0)	6 (33.3) 1 (5.6) 4 (22.2) 0 (0.0) 0 (0.0) 5 (27.8) 1 (5.6) 1 (5.6)
Ethnicity	Not Hispanic or Latino Hispanic or Latino Other Do not wish to answer Missing	3 (15.0) 10 (50.0) 1 (5.0) 4 (20.0) 2 (10.0)	8 (44.4) 6 (33.3) 1 (5.6) 0 (0.0) 3 (16.7)
Primary Language	English Spanish Missing	13 (65.0) 5 (25.0) 2 (10.0)	14 (77.8) 3 (16.7) 1 (5.6)
Highest Education Achieved *	Grade school Some high school High school diploma or GED Vocational degree Some college Associate degree Bachelor’s degree Graduate degree Do not wish to answer Missing	1 (5.0) 2 (10.0) 1 (5.0) 3 (15.0) 4 (20.0) 0 (0.0) 3 (15.0) 4 (20.0) 0 (0.0) 2 (10.0)	1 (5.6) 1 (5.6) 1 (5.6) 4 (22.2) 7 (38.9) 1 (5.6) 2 (11.1) 3 (16.7) 0 (0.0) 1 (5.6)
Relationship Status	Single, never married Married Living with someone as if married Widowed Divorced Separated Do not wish to answer Missing	2 (10.0) 9 (45.0) 4 (20.0) 0 (0.0) 1 (5.0) 2 (10.0) 0 (0.0) 2 (10.0)	1 (5.6) 14 77.8) 1 (5.6) 1 (5.6) 0 (0.0) 0 (0.0) 0 (0.0) 1 (5.6)
Household Income	Less than USD 10,000 USD 10,000 to USD 29,999 USD 30,000 to USD 49,999 USD 50,000 to USD 69,999 USD 70,000 to USD 89,999 USD 90,000 to USD 149,999 USD 150,000 or more Do not wish to answer Missing	1 (5.0) 4 (20.0) 1 (5.0) 1 (5.0) 0 (0.0) 5 (25.0) 2 (10.0) 3 (15.0) 3 (15.0)	0 (0.0) 6 (33.3) 1 (5.6) 3 (16.7) 0 (0.0) 3 (16.7) 4 (22.2) 0 (0.0) 1 (5.6)
Unable to pay rent/mortgage ^	Yes No Do not wish to answer Missing	7 (35.0) 10 (50.0) 1 (5.0) 2 (10.0)	3 (16.7) 15 (83.3) 0 (0.0) 0 (0.0)
Unable to pay utilities ^	Yes No Do not wish to answer Missing	7 (35.0) 9 (45.0) 2 (10.0) 2 (10.0)	1 (5.6) 16 (88.9) 0 (0.0) 1 (5.6)
Unable to seek medical care ^#,^^	Yes No Do not wish to answer Missing	2 (10.0) 15 (75.0) 1 (5.0) 2 (10.0)	1 (5.6) 16 (88.9) 0 (0.0) 1 (5.6)
Unable to seek dental care ^#,^^	Yes No Do not wish to answer Missing	41 (29.3) 90 (64.3) 1 (0.7) 8 (5.7)	3 (16.7) 15 (83.3) 0 (0.0) 0 (0.0)

Notes: * some parents selected more than one response, ^ in the past 12 months, ^#^ individual(s) in the household unable to seek care due to the inability to pay or lack of insurance.

**Table 3 cancers-17-00930-t003:** Caregiver psychological outcome scores, mean (SD).

Psychological Outcome Measure	Arm	Pre-Child HSCT	Day +60	Day +90	Day +180
BAI	BI	11.3 ± 9.6	5.3 ± 4.8	6.5 ± 3.8	6.7 ± 5.5
	Control	7.4 ± 8.4	9.5 ± 8.5	8.0 ± 8.1	8.2 ± 10.3
BDI	BI	14.3 ± 10.4	10.2 ± 8.5	8.5 ± 7.3	10.6 ± 8.6
	Control	9.6 ± 5.8	10.1 ± 10.3	9.7 ± 7.5	9.6 ± 9.6
SPSI	BI	13.2 ± 1.8	13.9 ± 2.0	15.0 ± 2.4	14.4 ± 2.0
	Control	13.6 ± 2.4	14.4 ± 2.6	14.0 ± 2.6	14.0 ± 2.9
IES	BI	29.0 ± 20.0	25.6 ± 13.5	29.5 ± 19.9	29.3 ± 12.4
	Control	29.0 ± 19.8	27.8 ± 18.4	24.6 ± 16.5	21.3 ± 19.5
POMS	BI	22.9 ± 25.5	15.1 ± 13.5	16.5 ± 15.3	24.2 ± 9.6
	Control	21.3 ± 19.7	23.6 ± 27.8	21.6 ± 24.7	15.1 ± 26.5

Notes: SD = standard deviation, BI = Bright IDEAS^®^, BAI = Beck Anxiety Inventory-I, BDI = Beck Depression Inventory-II, SPSI = Social Problem-Solving Skills Inventory Revised Short Form, IES = Impact of Events Scale Revised, POMS = Profile of Mood States Short Form.

**Table 4 cancers-17-00930-t004:** Caregiver anxiety and depression scores by clinical cutoff.

(**a**) Caregiver Anxiety Scores by Clinical Cutoff, N (%).						
**Clinical Cutoff**	**BAI-I Range**	**Arm**	**Pre-Child HSCT**	**Day +60**	**Day +90**	**Day +180**
Minimal	0–7	BI	6 (35.3)	9 (75.0)	6 (54.5)	6 (54.5)
		Control	11 (64.7)	6 (50.0	9 (60.0)	8 (66.7)
Mild	8–15	BI	5 (24.7)	2 (16.7)	5 (45.5)	5 (45.5)
		Control	3 (17.6)	4 (33.3)	4 (26.7)	2 (16.7)
Moderate	16–25	BI	5 (29.4)	1 (8.3)	0 (0.0)	0 (0.0)
		Control	2 (11.8)	1 (8.3)	2 (13.3)	1 (8.3)
Severe	26–63	BI	1 (5.9)	0 (0.0)	0 (0.0)	0 (0.0)
		Control	1 (5.9)	1 (8.3)	0 (0.0)	1 (8.3)
(**b**) Caregiver Depression Scores by Clinical Cutoff, N (%).						
**Clinical Cutoff**	**BDI-II Range**	**Arm**	**Pre-Child HSCT**	**Day +60**	**Day +90**	**Day +180**
Minimal	0–13	BI	6 (35.3)	6 (50.0)	7 (63.6)	5 (45.5)
		Control	12 (70.6)	6 (54.5)	9 (60.0)	8 (66.7)
Mild	14–19	BI	2 (11.8)	2 (16.7)	2 (18.2)	4 (36.4)
		Control	1 (5.9)	3 (27.3)	2 (13.3)	0 (0.0)
Moderate	20–28	BI	5 (29.4)	3 (25.0)	0 (0.0)	2 (18.2)
		Control	2 (11.8)	0 (0.0)	2 (13.3)	1 (8.3)
Severe	29–63	BI	4 (23.5)	1 (8.3)	2 (18.2)	0 (0.0)
		Control	2 (11.8)	2 (18.2)	2 (13.3)	3 (25.0)

Notes: BI = Bright IDEAS^®^, BAI = Beck Anxiety Inventory, BDI = Beck Depression Inventory.

**Table 5 cancers-17-00930-t005:** Qualitative themes from semi-structured feedback interviews (*N* = 7).

Acceptability		
Theme	Definition/Subthemes	Example
Satisfaction	Positive experience; appreciation; would recommend to others	“I found it very resourceful to problem solve and navigate certain issues and emotions. So in general, I really liked it and felt very appreciative about it.” “I feel like this should be offered, because I’m so glad I did it, because, you know, it really helped me…I just feel as though it’s a really, really good program.”
Exceeded Expectations	Gladness to have participated in BI despite an initial reticence to enroll	“So it seemed like I had a lot in my plate, and I was doubtful about participating in this program. But it helped.” “I almost said no because I was like…I have so much going on. But I’m glad I said ‘yes’.”
**Feasibility**		
**Theme**	**Definition/Subthemes**	**Example**
Barriers	Time; interruptions; childcare	“There were several times that we started and it just didn’t work because the baby was very, very fussy or there was a lot going on.”
Facilitators	Flexible scheduling; participant choice	“[The interventionist] was understanding and would wait for me.” “I was given the option to, at any point, request more visits or space them out, and I just felt for me once a week was a good fit.”
**Implementation**		
**Theme**	**Definition/Subthemes**	**Example**
Format	Feedback regarding delivery format (in person vs. telehealth)	“Applying pro and con, it is sometimes better in person because it is easier to express ideas, express feelings face to face.” “I didn’t mind the Telehealth because it was wonderful to have that flexibility. But me personally, I like it better in person.”
Timing of intervention	Feedback regarding timing or number of BI sessions	“Starting it before that [transplant] process probably would have helped make the process, like, easier. Instead of saying okay, we’re at this critical point and now we’re going to start this...don’t let it get to the critical point.” “[If] you eventually went to offer it to me before, that would have also been very overwhelming.” “I was comfortable with [the BI model] by the time I came home…I don’t know that I needed the follow up…I’m ready to, like, fly and do this on my own.”
Use of Worksheets	Feedback regarding BI worksheets	“And even writing down other solutions was very interesting because you could see the amount of solutions this problem has and just get the best one in the moment.”
Role of BI Interventionist	Active listening; support	“I felt like all over the place sometimes, and then [the interventionist] would kind of like, on the head, like this is what I hear.” “And it felt like I had someone to rely on when I was alone or feeling down, or you know, overstressed.”
	Guiding BI skill use	“Having someone guide me through the thought process and have a plan to deal with my issues, that helped me.” “I tend to overthink, so…it was helpful to have someone kind of simplify things.”

## Data Availability

The data presented in this study are available on request from the corresponding author due to the inclusion of limited identifying information within the dataset.

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
