# Peer review of "Problem-Solving Skills Training for Parents of Children Undergoing Hematopoietic Stem Cell Transplantation: A Mixed Methods Feasibility Study"

_cancers, 2025, doi:10.3390/cancers17060930_

Round 1
Reviewer 1 Report
Comments and Suggestions for Authors
This paper describes the feasibility and acceptability, as well as preliminary efficacy, of an evidence-based intervention (Bright Ideas) for caregivers of children undergoing hematopoietic stem cell transplant (HSCT). The paper is well written and nicely summarizes the objectives and results, without overstating conclusions. I believe this study would make a positive contribution to the literature and have only minor suggestions as noted below.
Minor edit - P. 5, Line 232: The Impact of Events Scale-Revised; authors should note what higher or lower scores indicate to help reader interpret results. Also recommend then moving the Profile of Mood States paragraph into a new paragraph for consistency with prior sections.
Minor edit – P. 6. Line 24: change second 2.6 subheading to 2.7
Section 2.4 – I would like more information regarding logistics of intervention delivery, given the setting. It would be helpful to clarify the extent to which (if not all) sessions took place during the HSCT hospital admission, and if so if they were conducted in or out of the patient room. How were discharges handled and were sessions routinely rescheduled? This information is particularly important to describe given the goal to examine feasibility; to generalize this intervention others would need to understand how scheduling of caregiver availability and room space etc were determined as this can be challenging in the inpatient setting.
Additional information about treatment course would also be helpful in the context of a larger study. HSCT is a high morbidity and mortality process, and this would likely impact parent functioning a great dela. Controlling for the child’s well-being statistically will be important in any larger studies.
Finally, it would be helpful to include a measure of current medical (or even caregiving) stress for caregivers. The IES-R is the only measure that really captures parent perceptions of a stressful event, but this is phased retrospectively despite the fact that in this study they are in the midst of the stressful situation. A measure more fine tuned to the experience of parents during HSCT would be helpful.
The authors address the biggest challenge with the conceptual basis of this paper in their limitations section; that is, the number of participants who declined in addition to the number of BI participants who withdrew call into question the possibility of selection bias with responses.
Author Response
Response to Reviewers for Cancers-3482046
On behalf of our team, I want to extend our gratitude to the reviewers for their time in reviewing this manuscript and providing important comments that will strengthen the manuscript. We agree with all reviewer suggestions and have attempted to thoroughly address each comment; please see below for detailed item-by-item response to each reviewer. For ease in reviewing, we have highlighted the location in the manuscript of each requested revision.
In addition to uploading the clean revised manuscript, we have also included a tracked-changes version of the edited manuscript as a supplementary file to assist in highlighting changes that were made.
Please let us know if you have any questions.
Author Responses
Reviewer 1 |
Author Responses |
Minor edit - P. 5, Line 232: The Impact of Events Scale-Revised; authors should note what higher or lower scores indicate to help reader interpret results. Also recommend then moving the Profile of Mood States paragraph into a new paragraph for consistency with prior sections. |
Revised accordingly |
Minor edit – P. 6. Line 24: change second 2.6 subheading to 2.7 |
Revised accordingly
|
Section 2.4 – I would like more information regarding logistics of intervention delivery, given the setting. It would be helpful to clarify the extent to which (if not all) sessions took place during the HSCT hospital admission, and if so if they were conducted in or out of the patient room. How were discharges handled and were sessions routinely rescheduled? This information is particularly important to describe given the goal to examine feasibility; to generalize this intervention others would need to understand how scheduling of caregiver availability and room space etc were determined as this can be challenging in the inpatient setting. |
We have clarified this information in section 2.4. We have also added information to section 3.1 regarding the percentage of in person versus telehealth sessions that took place.
|
Additional information about treatment course would also be helpful in the context of a larger study. HSCT is a high morbidity and mortality process, and this would likely impact parent functioning a great deal. Controlling for the child’s well-being statistically will be important in any larger studies. |
Thank you for this insight. These concepts were added to the future research section. |
It would be helpful to include a measure of current medical (or even caregiving) stress for caregivers. The IES-R is the only measure that really captures parent perceptions of a stressful event, but this is phased retrospectively despite the fact that in this study they are in the midst of the stressful situation. A measure more fine tuned to the experience of parents during HSCT would be helpful. |
Thank you for this insight. These concepts were added to the future research section. |
The authors address the biggest challenge with the conceptual basis of this paper in their limitations section; that is, the number of participants who declined in addition to the number of BI participants who withdrew call into question the possibility of selection bias with responses. |
The authors appreciate this insight. |
Reviewer 2 Report
Comments and Suggestions for Authors
The manuscript is long and should be shortened.
Comments on the Quality of English LanguageWeak and redundant sentences are utilized in several parts of the paper; rephrasing is required.
Author Response
Response to Reviewers for Cancers-3482046
On behalf of our team, I want to extend our gratitude to the reviewers for their time in reviewing this manuscript and providing important comments that will strengthen the manuscript. We agree with all reviewer suggestions and have attempted to thoroughly address each comment; please see below for detailed item-by-item response to each reviewer. For ease in reviewing, we have highlighted the location in the manuscript of each requested revision.
In addition to uploading the clean revised manuscript, we have also included a tracked-changes version of the edited manuscript as a supplementary file to assist in highlighting changes that were made.
Please let us know if you have any questions.
Reviewer 2 |
|
The manuscript is long and should be shortened. |
Revised accordingly to shorten where appropriate throughout. |
Weak and redundant sentences are utilized in several parts of the paper; rephrasing is required. |
Weak and redundant sentences have been removed, revised or rephrased. |
|
|
Reviewer 3 Report
Comments and Suggestions for Authors
Thank you for the opportunity to review your manuscript . Overall, it is clear, well-written, and clinically relevant. I believe this manuscript has the potential to contribute to the literature; however, several concerns must be addressed to enhance its overall utility. Major points of concern are detailed below.
Introduction
- Please include a table or figure to further demonstrate/describe Bright IDEAS intervention session content.
- On page 3, the authors mention that they used a mixed-method approach to evaluate feasibility and acceptability, incorporating caregivers' feedback to identify modifiable factors for future iterations of the intervention. This is somewhat confusing since modifiable factors can represent potential targets of intervention. Instead, authors can state that qualitative feedback can be used to further refine intervention content, features, and study procedures.
Methods
- Authors state that intervention participants attended up to 8 sessions. How were the number of sessions each participant received determined?
- The control condition title, “standard psychosocial care,” is somewhat confusing, especially because psychosocial supports are available for both groups. Please refer to this control condition as “treatment as usual” or “usual care.” More information on control conditions used in RCTs can be found in: Freedland, K.E., Mohr, D.C., Davidson, K.W., Schwartz, J.E. Usual and unusual care: existing practice control groups in randomized controlled trials of behavioral interventions. Psychosomatic Medicine 2011, 73(4), 323-35.
- Similarly, given the sample size of the current sample, psychosocial support interactions could not be accounted for, which may have influenced the results. Please add this as a study limitation/future direction.
- Given that not all children receiving HSCT have a cancer diagnosis, was the Social Problem-Solving Skills Inventory-Cancer modified in any way to be applicable to caregivers of children without cancer? Providing more detailed information about this measure in the method section may clarify concerns related to this comment.
- Please provide a rationale for conducting semi-structured interviews with only half of the caregivers who completed the Bright IDEAS intervention. Why did authors not include all caregivers? Where there any apparent differences between participants who completed the interviews and those who did not in terms of demographics, child diagnosis, language, or other factors?
- Please elaborate on the methods used to analyze qualitative data. Specifically, please provide details on the coding process, including whether dual coding methods were employed, if intercoder agreement was assessed, and how themes were generated.
Results
- It is recommended for authors to use the term “white” in place of “Caucasian.” Caucasian originates from a racial classification system that is considered outdated. Refer to the following publication for more information: Shamambo, L.J., Henry, T.L. Rethinking the Use of "Caucasian" in Clinical Language and Curricula: A Trainee's Call to Action. J Gen Intern Med, 2022, 37(7), 1780-1782.
- How many Spanish-speaking caregivers were in each group?
- The intervention group had a higher number of caregivers of children with a cancer diagnosis. These caregivers reported higher levels of anxiety and depression at baseline compared to the control group, indicating they may experience more psychological distress. This finding aligns with existing literature on caregiver distress in pediatric cancer. The authors should acknowledge this as a limitation of the study and discuss its potential implications.
- On page 9, the authors state, “More caregivers in the control group reported severe depression at day 180 compared to baseline.” However, it appears that only one additional caregiver reported severe depression (3 versus 2 at previous time points), so the term “more” may be seen as an exaggeration.
Discussion
- On page 12, authors report that the Bright IDEAS intervention was deemed feasible and acceptable in this population; however, no conditions were reported for how feasibility was determined (e.g., “the intervention would be deemed feasible if the following conditions were met: XX% enrollment rate, XX% retention rate, XX% data completion rate). Given the somewhat low enrollment rate (59%), further refinement of study procedures may be needed prior to efficacy testing.
- Similarly, on page 13, authors state that target enrollment rates were not met. Enrollment rate expectation was not reported.
- Also on page 12, authors state that no adverse events were reported. Details regarding adverse event reporting are missing from the Results and Methods sections.
- Similar to comment #2 in Results above, authors acknowledge that higher levels of psychological distress were reported by caregivers in the intervention group at baseline (page 12). Could this be related to child diagnosis?
- Please include a description or citation that details how Bright IDEAS intervention and study materials were translated into Spanish.
Author Response
Response to Reviewers for Cancers-3482046
On behalf of our team, I want to extend our gratitude to the reviewers for their time in reviewing this manuscript and providing important comments that will strengthen the manuscript. We agree with all reviewer suggestions and have attempted to thoroughly address each comment; please see below for detailed item-by-item response to each reviewer. For ease in reviewing, we have highlighted the location in the manuscript of each requested revision.
In addition to uploading the clean revised manuscript, we have also included a tracked-changes version of the edited manuscript as a supplementary file to assist in highlighting changes that were made.
Please let us know if you have any questions.
Reviewer 3 |
|
Introduction |
|
1. Please include a table or figure to further demonstrate/describe Bright IDEAS intervention session content. |
A table has been added to section 2.4 to assist in further describing Bright IDEAS session content. We have additionally included references in the introduction in which the intervention is described. |
2. On page 3, the authors mention that they used a mixed-method approach to evaluate feasibility and acceptability, incorporating caregivers' feedback to identify modifiable factors for future iterations of the intervention. This is somewhat confusing since modifiable factors can represent potential targets of intervention. Instead, authors can state that qualitative feedback can be used to further refine intervention content, features, and study procedures. |
Thank you for this feedback. This statement has been revised. |
Methods |
|
1. Authors state that intervention participants attended up to 8 sessions. How were the number of sessions each participant received determined? |
We have clarified this determination in the methods section, as well as its consistency with prior research utilizing this manualized intervention. |
2. The control condition title, “standard psychosocial care,” is somewhat confusing, especially because psychosocial supports are available for both groups. Please refer to this control condition as “treatment as usual” or “usual care.” More information on control conditions used in RCTs can be found in: Freedland, K.E., Mohr, D.C., Davidson, K.W., Schwartz, J.E. Usual and unusual care: existing practice control groups in randomized controlled trials of behavioral interventions. Psychosomatic Medicine 2011, 73(4), 323-35. |
Thank you for this reference. Control condition language has been revised throughout. |
3. Similarly, given the sample size of the current sample, psychosocial support interactions could not be accounted for, which may have influenced the results. Please add this as a study limitation/future direction. |
Assessment of caregiver psychosocial support was added to the future research section. |
4. Given that not all children receiving HSCT have a cancer diagnosis, was the Social Problem-Solving Skills Inventory-Cancer modified in any way to be applicable to caregivers of children without cancer? Providing more detailed information about this measure in the method section may clarify concerns related to this comment. |
Thank you for your attention to this. Our inclusion of “Cancer” in this measure’s title was an error in the manuscript text. The text has been revised to correctly sate that the instrument used was the Social-Problem-Solving Skills Inventory-Revised-Short-Form. The measure is not disease-specific; therefore, no modifications were needed. |
5. Please provide a rationale for conducting semi-structured interviews with only half of the caregivers who completed the Bright IDEAS intervention. Why did authors not include all caregivers? Where there any apparent differences between participants who completed the interviews and those who did not in terms of demographics, child diagnosis, language, or other factors? |
The rationale and requested information have been added to sections 2.6 and 3.2. |
6. Please elaborate on the methods used to analyze qualitative data. Specifically, please provide details on the coding process, including whether dual coding methods were employed, if intercoder agreement was assessed, and how themes were generated. |
Revised accordingly in section 2.7. |
Results |
|
1. It is recommended for authors to use the term “white” in place of “Caucasian.” Caucasian originates from a racial classification system that is considered outdated. Refer to the following publication for more information: Shamambo, L.J., Henry, T.L. Rethinking the Use of "Caucasian" in Clinical Language and Curricula: A Trainee's Call to Action. J Gen Intern Med, 2022, 37(7), 1780-1782. |
Revised accordingly. Thank you for the reference. |
2. How many Spanish-speaking caregivers were in each group? |
This information is included in the Sample Characteristics Table (previously Table 1, now Table 2). |
3. The intervention group had a higher number of caregivers of children with a cancer diagnosis. These caregivers reported higher levels of anxiety and depression at baseline compared to the control group, indicating they may experience more psychological distress. This finding aligns with existing literature on caregiver distress in pediatric cancer. The authors should acknowledge this as a limitation of the study and discuss its potential implications. |
Additional text was added to section 4.2. |
4. On page 9, the authors state, “More caregivers in the control group reported severe depression at day 180 compared to baseline.” However, it appears that only one additional caregiver reported severe depression (3 versus 2 at previous time points), so the term “more” may be seen as an exaggeration. |
This statement was removed. |
Discussion |
|
1. On page 12, authors report that the Bright IDEAS intervention was deemed feasible and acceptable in this population; however, no conditions were reported for how feasibility was determined (e.g., “the intervention would be deemed feasible if the following conditions were met: XX% enrollment rate, XX% retention rate, XX% data completion rate). Given the somewhat low enrollment rate (59%), further refinement of study procedures may be needed prior to efficacy testing.
Similarly, on page 13, authors state that target enrollment rates were not met. Enrollment rate expectation was not reported. |
The discussion was revised to clarify the interpretation of the present study’s enrollment rate, and to clarify relevant limitations and further refinement needs for future research.
The target enrollment was added to this section (4.2) for clarity, in alignment with the target enrollment reported in section 2.2. |
2. Also on page 12, authors state that no adverse events were reported. Details regarding adverse event reporting are missing from the Results and Methods sections. |
This statement was removed as there was no formal protocol for adverse event reporting. |
3. Similar to comment #2 in Results above, authors acknowledge that levels of psychological distress were reported by caregivers in the intervention group at baseline (page 12). Could this be related to child diagnosis? |
Thank you for this feedback. This was further clarified and discussed in section 4.2. |
4. Please include a description or citation that details how Bright IDEAS intervention and study materials were translated into Spanish. |
The relevant citation was added in section 2.4. |
Author Edits |
Updated references Updated Table #s Added MDPI requests: Author affiliations; Simple summary; Author contributions; Funding; Institutional Review Board statement; Informed consent statement; Data availability statement; Conflicts of interest |
Thank you again for your reviews, which we believe have significantly strengthened the manuscript. We will be happy to address any further questions or concerns.
Round 2
Reviewer 3 Report
Comments and Suggestions for Authors
Thank you for the opportunity to review your revised manuscript. I have just one additional suggestion. Please include as a limitation that all caregivers who participated in the semi-structured interviews had a child with a malignant diagnosis and received an allogeneic transplant. Additionally, discuss the potential implications of this limitation and suggest areas for future research.
